# The experience of being a mother with end stage renal disease: A qualitative study of women receiving treatment at an ambulatory dialysis unit

Miriam Álvarez-Villarreal[1], Juan Francisco Velarde-García[2]*, Cristina García-Bravo[3], Pilar Carrasco-Garrido[4], Carmen Jimenez-Antona[5], Paloma Moro-Lopez-Menchero[5], Domingo Palacios-Ceña[5]

1 Dialysis Unit, Hospital Universitario Infanta Cristina, Parla, Spain, 2 Department of Nursing, Spanish Red Cross, Instituto de Investigación Sanitaria Gregorio Marañón (iiSGM), Universidad Autonoma de Madrid, Madrid, Spain, 3 Department of Physical Therapy, Occupational Therapy, Physical Medicine and Rehabilitation, Research Group in Evaluation and Assessment of Capacity, Functionality and Disability of Universidad Rey Juan Carlos (TO+IDI), Alcorcón, Spain, 4 Department of Medical Specialties and Public Health, Universidad Rey Juan Carlos, Alcorcón, Spain, 5 Department of Physical Therapy, Occupational Therapy, Physical Medicine and Rehabilitation, Research Group of Humanities and Qualitative Research in Health Science of Universidad Rey Juan Carlos (Hum&QRinHS), Alcorcón, Spain

☯ These authors contributed equally to this work.
* jvg@cruzroja.es

**Citation:** Álvarez-Villarreal M, Velarde-García JF, García-Bravo C, Carrasco-Garrido P, Jimenez-Antona C, Moro-Lopez-Menchero P, et al. (2021) The experience of being a mother with end stage renal disease: A qualitative study of women receiving treatment at an ambulatory dialysis unit. PLoS ONE 16(9): e0257691. https://doi.org/10.1371/journal.pone.0257691

**Data Availability Statement:** Data may be obtained from a third party and are not publicly

## Abstract

### Background

End-stage kidney disease (ESKD) has considerable effects on the quality of life, impairing daily activities and leading to lifestyle changes. The purpose of this study was therefore to explore the experience of motherhood and taking care of children in women with ESKD.

### Methods

A qualitative exploratory study was conducted based on an interpretive framework. Participants were recruited using non-probabilistic purposeful sampling. In total, 14 women with ESKD were included, who were treated at the dialysis unit of a Spanish hospital. In-depth interviews (unstructured and semi-structured interviews) and researchers' field notes were used to collect the data. A systematic text condensation analysis was performed. The techniques performed and application procedures used to control trustworthiness were credibility, transferability, dependability, and confirmability.

### Results

Three themes emerged from the data. "Coping with being a mother" described how women are faced with the decision to become mothers and assess the risks of pregnancy. The second theme, called "Children and the experience of illness", highlighted the women's struggle to prevent the disease from affecting their children emotionally or disrupting their lives. The

available. All data relevant to the study are included in the article or uploaded as online supplemental information.

**Funding:** This article has not received funding by Public or Private Institutions. We have no financial interests and we have not received direct or indirect funding.

**Competing interests:** Any potential conflicts of interest are disclosed. All individuals named as authors qualify for authorship. All persons listed as authors have participated sufficiently in the work to take public responsibility for the content of the manuscript and have approved the final version.

third theme, "Fear of genetic transmission", was based on the women's fear of passing the disease on to their children.

## Conclusions

Deciding to become a mother and taking care of children represents a challenge for women with ESKD, coupled with the losses in their lives caused by the disease. These findings are only relevant to women on dialysis.

## Introduction

Chronic kidney disease (CKD) is defined as the presence of alterations in renal structure or function for at least three months, causing numerous health implications [1]. The diagnosis of CKD is based on markers of kidney damage or a reduction in estimated glomerular filtration rate (eGFR) below 60 ml/min/1.73 m$^2$ [2].

A significant percentage of the population suffer from CKD, which is related to highly prevalent phenomena or diseases, such as ageing, arterial hypertension, diabetes, cardiovascular disease, autoimmune disease or genetic renal disease [3]. The global prevalence of CKD is around 60–80 cases per 100,000 individuals in Europe and the USA. It is estimated that more than 10% of the world's population suffers from some degree of CKD [4]. CKD is an irreversible and progressive disease. Once the lesion occurs, renal function progressively deteriorates with the likelihood of progressing to functional loss [5]. The most severe manifestation of CKD is ESKD, which has been increasing in incidence and prevalence for the past two decades [6].

Once the kidneys fail, it is necessary to start renal replacement therapy (RRT) to help replace their functions. Dialysis is an artificial therapeutic procedure for the removal of toxic substances in the blood and excess water [7]. Several modalities of dialysis exist; hemodialysis (HD) performed at a hospital or dialysis center, home HD and peritoneal dialysis (PD). The latter two are more flexible than HD as they can be adjusted to the patient's schedule and activities. Patients who receive home dialysis only have to go to the hospital every 1 or 2 months for check-ups. Furthermore, the equipment the individual needs is delivered regularly to the home, whereas in the case of HD, the patient has to adapt to the schedules established by the hospital and has to travel to undergo each treatment [7]. Renal transplantation allows for RRT withdrawal. and enables the patient's physiology to more closely return to the pre-ESKD state [8]. The duration with RRT until transplantation is variable, between 3 and 5 years [9].

In women, ESKD has profound implications for global health and constitute a significant and growing proportion of the dialysis population worldwide [10, 11]. End stage renal disease can affect women during their most fertile years, aggravating the burden of their disease in many cases due to the devastating loss of pregnancy. Thus, pregnancy becomes a difficult prospect for these women [12]. The inability to achieve a successful pregnancy is deeply traumatic for women with ESKD and has detrimental consequences on their identity, relationships, and emotional well-being [13]. Decision-making in relation to future pregnancy requires women to face uncertainties about their own survival, disease progression, infant health outcomes and genetic transmission [14]. In addition, women feel a responsibility towards the health of their partners, children, and other family members [15]. Maternity is associated with several common fears, including the transmission of hereditary kidney disease to children and the inability to survive or be well enough to care for children

[16]. In addition to managing their RRT, women with ESKD under dialysis treatment try to focus on fitting everything in, maintaining ties between family members, remaining present in the lives of their children and partner, and not losing the connection to their family and social life, and striving for normality [17].

The experience of having kidney disease is highly individual, and qualitative research can provide a more holistic view that may be more meaningful to health care professionals [18]. Previously, qualitative studies have been used to describe the beliefs, values, and experiences of pregnant women with advanced stages of CKD [19]. Also, further qualitative, and quantitative studies are needed to support the provision of counselling for patients, families, and health professionals on this topic [20]. The purpose of this study was therefore to explore the experience of motherhood and taking care of children among women with ESKD.

## Materials and methods

### Study design

A qualitative exploratory study was conducted based on an interpretive framework [18]. This study was conducted according to the Standards for Reporting Qualitative Research (SRQR) [21].

### Ethics

The study was approved by the Local Ethical Committee of Universidad Rey Juan Carlos (code: 2806201711017), and the Clinical Research Ethics Committee at Hospital Universitario Puerta de Hierro Majadahonda (code:11.17). All participants provided oral informed consent prior to their inclusion.

### Research team

Prior to the study, the researchers' positioning was established via two briefing sessions addressing the theoretical framework for this qualitative study, their beliefs, and their motivation for the research [18]. Seven researchers (five women) participated in this study, including three nurses (MAV, JFVG, DPC), one occupational therapist (CGB), one pharmacologist (PCG), and two physiotherapists (CJA, PMLM). All researchers had experience in research in health sciences.

### Participants, context, and sampling strategies

The study included women with CKD attending the Ambulatory Dialysis Unit at the hospital belonging to the public health system of Madrid (Spain). The inclusion criteria were: (a) women, (b) over the age of 18, (c) with ESKD, d) diagnosed with CKD stage 5 (kidney failure-FGR < 15 ml/min/1,73 m2), following the criteria of Kidney Disease Improving Global Outcomes (KDIGO) [1, 22]. and (e) women who are receiving or have received RRT. In the present study, the inclusion criteria included women who required RRT and who finally received renal transplantation. The exclusion criteria were: (a) acute kidney injury (AKI) requiring HD, (b) serious psychiatric or cognitive disorders, (c) inability to communicate in Spanish or provide informed consent.

Purposive sampling was used, based on relevance to the research question (not clinical representativeness) [18]. Sampling and data collection were pursued until the researchers achieved information redundancy, at which point no new information emerged from the data analysis [18].

## Data collection

Data were collected over a seven-month period between October 2017 and April 2018. After collecting professional and personal data from each woman, the first stage of data collection consisted of unstructured interviews, using open questions, such as: What is your experience with CKD and being a mother and taking care of your children? The second stage consisted of semi-structured interviews to obtain information regarding specific topics of interest [18]. The questions were developed based on accounts obtained from the women. Open-ended follow-up questions were also used to obtain the detailed descriptions, including the following: 1, "Do you consider that ESKD has changed your life?"; 2, " How was it when you were diagnosed with ESKD, what was the most relevant issue for you?"; 3, "What challenges have you faced being a mother and/or caring for your children having ESKD?"; 4, "What are the most relevant changes that have taken place in your family life?"; 5, "Do you think the disease has impacted your family, and your children?, If so, how?"; 6, Can you describe your emotional experience of being a mother and having ESKD in a single sentence? Additionally, "Please tell me more about that", was also used during all the interviews (if needed) to enhance the depth of the discussion of a specific topic. During the interview, at the women's request, it was clarified that when they were asked about the care of their children during RRT, this included both the care provided to the infants and the daily care that the woman provided to the remaining children she already had.

The interviews were conducted by MAV, DPC, and JVG. The interviews were audio-recorded and transcribed verbatim. A total of 14 interviews were conducted (one per woman). Overall, 929 min of interviews were recorded, with 348 min corresponding to the first stage and 581 min corresponding to the second stage. Each of the first-stage interviews lasted between 20 and 40 min (mean duration: 30 min), whereas the second stage interviews lasted between 25 and 60 min (mean duration: 42.5 min). All interviews were conducted at the women's homes or in a private hospital room, according to the women's preference. Fourteen researcher field notes were also collected. The researcher field notes provided a rich source of information as participants described their personal experiences, their behavior during data collection, and enabled them to note their reflections concerning methodological aspects of the data collection [18]. During data collection nobody else was present besides the participants.

## Data analysis

Complete verbatim transcripts were produced for each of the interviews. The analysis was conducted by MAV, DPC, and JVG. The initial results were subsequently merged in joint sessions, during which data collection and analysis procedures were discussed. In the case of differences of opinion, theme identification was decided by consensus. A systematic text condensation analysis was performed [23].

Systematic text condensation (STC) is an elaboration of Giorgi's principles (follower of Husserl) [24], including four comparable steps of analysis. In this descriptive approach, presenting the experience of the participants as expressed by themselves, and following Giorgi, STC represents an explorative proposal to present vital examples from peoples' life worlds [23]. Also, like Giorgi's method, STC implies analytic reduction with decontextualization and recontextualization of data. The procedure consists of the following steps [23]: 1) Total impression; reading all the material and an overview of the data. 2) Identifying and sorting meaning units and codes; systematically reviewing the transcript line by line to identify meaning units. A meaning unit is a text fragment containing some information about the research question. Subsequently coding begins (decontextualization), which includes identifying, classifying, and sorting meaning units and marking these with a code—a label that connects related meaning

units into a code group. 3) Condensation–from code to meaning; implying the systematic abstraction of meaning units within each of the code groups established in the second step of analysis. Empirical data are reduced to a decontextualized selection of meaning units sorted as thematic code groups across individual participants. 4) Synthesizing–from condensation to descriptions and concepts; data are reconceptualized, putting the pieces together again. By synthesizing the contents of the condensates, descriptions and concepts are developed, providing stories that reflect the participants' experiences [23]. Each of the interviews were analyzed separately, without performing any comparison between one and the other. After analyzing each interview, each researcher listed their themes and confluent and diverging issues were negotiated. Finally, in the case of different opinions, theme identification was decided by consensus. No qualitative software was used on the data.

### Rigor

The techniques performed and application procedures used to control trustworthiness are described in (Table 1) [18].

## Results

Fourteen women with CKD were recruited. The mean age of women was 57,5 years (standard deviation, SD: 12,30), the median time from the beginning of RRT after diagnosis was 6 years (SD: 10,21). Also, regarding treatment type, 2 participants (14,3%) had peritoneal dialysis, 1 participant (7,1%) had kidney transplant, and 11 participants (78,6%) had hemodialysis. Moreover, 12 women had a partner. The mean number of children was 2,28 (SD: 1,53). See Table 2.

Three specific themes emerged from the data analysed: a) Coping with being a mother, b) having children and having a chronic disease, and c) Fear of genetic transmission. We included some of the women's narratives taken directly from the interviews in relation to the emerging themes.

### Coping with being a mother

For women who have children or wish to become pregnant, having CKD means facing difficulties and making life-changing decisions. The women interviewed who already had children

**Table 1. Trustworthiness criteria.**

| Criteria | Techniques Performed and Application Procedures |
|---|---|
| Credibility | Investigator triangulation: each interview was analyzed by three researchers. Team meetings were performed in which the analyses were compared, and categories and themes were identified. |
| | Triangulation of methods of data collection: unstructured, semistructured interviews were conducted and researcher field notes were kept. |
| | Participant validation: asking the participants to confirm the data obtained at the stages of data collection. All participants were offered the opportunity to review the audio and/or video records to confirm their experience. None of the participants made additional comments. |
| Transferability | In-depth descriptions of the study performed, providing details of the characteristics of researchers, participants, contexts, sampling strategies, and the data collection and analysis procedures. |
| Dependability | Audit by an external researcher: an external researcher assessed the research protocol, focusing on aspects concerning the methods applied and study design. An external researcher specifically checked the description of the coding tree, the major themes, participants' quotations, quotations' identification, and themes' descriptions. |
| Confirmability | Investigator triangulation, participant triangulation, and data collection triangulation. |
| | Researcher reflexivity was encouraged via the performance of reflexive reports and by describing the rationale behind the study. |

**Table 2. Data on renal replacement therapy, and age of the children at the start of treatment.**

| Participant | Age | Type RRT[a] | Initial RRT | Time since CKD[e] diagnosis until beginning of RRT (years) | n° offspring | Age of children at start of treatment (years) | Children who lived in the home of the woman with ESKD[f] | Partner |
|---|---|---|---|---|---|---|---|---|
| 1 | 63 | HD[b] | 1993 | < 1 | 1 | 11 | Yes | No. Separated |
| 2 | 33 | HD | 2011 | 3 | 2 | 1 and 3 | Yes | Yes |
| 3 | 67 | HD | 2001 | 7 | 4 | Not available | Not available | Yes |
| 4 | 54 | HD | 2014 | < 1 | 6 | 13, 18, 23, 26, 32 and 34 | Yes | Yes |
| 5 | 64 | HD | 2015 | < 1 | 2 | 39 and 42 | No | No |
| 6 | 47 | PD[c] | 2014 | 9 | 2 | 13 and 15 | Yes | Yes |
| 7 | 50 | HD | 2017 | 5 | 1 | 21 | Yes | Yes |
| 8 | 56 | PD | 2017 | 16 | 1 | 30 | Yes | No. Separated |
| 9 | 64 | TxR[d] | 2015 | 5 | 2 | 34 and 38 | No | Yes |
| 10 | 46 | HD | 2017 | < 1 | 2 | 21 and 23 | Yes | Yes |
| 11 | 69 | HD | 2000 | < 1 | 5 | 15, 25, 27, 29 and 31 | Yes | Yes |
| 12 | 72 | HD | 2014 | < 1 | 1 | 42 | No | Yes |
| 13 | 42 | HD | 2017 | < 1 | 1 | 16 | Yes | Yes |
| 14 | 78 | HD | 2017 | 39 | 2 | Not available | Not available | Yes |

[a]RRT: renal replacement therapy;

[b]HD: hemodialysis;

[c]PD: peritoneal dialysis;

[d]TxR: kidney transplant;

[e]CKD: chronic kidney disease;

[f]ESKD: end-stage kidney disease

before starting RRT no longer considered having more children to be an option once they began treatment:

> *"When the disease appeared, the first thing it affected was my menstruation. Any desire to have more children or get pregnant disappeared. I had to assume that the daughters I had are the only ones I could have in this life."* (P4)

> *"Not as a result of the illness, no. On the contrary, I asked God not to get pregnant. With this illness there would be no way, I wouldn't know how to cope."* (P6).

The women described how receiving the news from the doctor that it would be difficult to become pregnant and that the disease posed a risk to the fetus was the worst part of the illness. For some of them, it even caused problems in their relationship with their partner:

> *"It was the worst thing about the disease. We really wanted to have children again, but it was inevitable, there was a lot of risk at stake. My husband, when I told him, told me straight out that if he had known, he would never have married me."* (P1).

Against medical advice, some women reported that their desire to become mothers was so strong that they risked their own health to try and become pregnant:

> *"When you want to have a baby you have to be disciplined, that was my challenge, I defied nature, I defied God because I longed to be a mother. I had the opportunity to be a mother. I*

*am thankful because my children are a gift, I wasn't obedient, but I had my children and I am happy."* (P2).

Some women who had been on RRT (PD) for a short time, acknowledged that they did not consider motherhood, although they would like to become mothers once again in the future. They reported that they had not asked or discussed this option during medical consultations and were unaware they could or should become pregnant:

*"I wouldn't get pregnant right now, but I would like to get pregnant in the future again."* (P6).

*"I hadn't thought about it, well I thought that at some point I could be a mum again. But now with everything I am going through, I have to say no".* (P8).

*"To tell you the truth, I don't really know if I can or if I should. I have never asked the doctor."* (P3).

## Children and the experience of illness

Our participants acknowledged that having children conditioned their experience of the disease and RRT. These women strived to ensure that their illness and its treatment would not affect their children's lives in any way. They tried not to let their children see them when they were physically unwell, sad, or downcast. They tried to spare them of any suffering from the illness at all costs. This involved not complaining in front of them, continuing to do their homework, accompanying them to their activities, picking them up from school, despite being exhausted or feeling sick from the treatment:

*"My day-to-day life is focused on my children. At least I don't have time to get depressed."* (P2).

*"My daughter would come home from school, and I still went down to pick her up many days when I was unwell, especially after dialysis. I tried to avoid getting her involved. I would put on my heels, put on my lipstick, comb my hair and even if I was a wreck inside, and I couldn't cope with my life, I would go and pick her up, so that she wouldn't see that I was unwell. I couldn't get depressed. I was mentally determined that my little girl didn't have to suffer for anything."* (P1).

*"Before they could see me bitter, I would tell my husband to take them out, even if I missed out on things. I didn't want suffering or sadness to be a part of their memories. I wanted to replace these with joy, with good memories. They have that memory since they were little, of seeing mum hooked up to machines with IVs."* (P5).

Many women spoke of the difficulties trying to preserve the same family dynamics from prior to beginning dialysis treatment. This represented one of the hardest things about the disease, associated with feelings of grief and pain:

*"I came out of dialysis feeling very poorly and I couldn't be with my daughter, accompany her and share the most ordinary things in life with her. When I was diagnosed with the disease, I was thinking about my daughter, who was 7 or 8 years old at the time. On a personal level it affects you a lot."* (P9).

*"I knew what was going to happen [as a result of the illness] with my daughter, I felt very sorry for her. My daughter has been brought up like a suitcase, here I leave you, here I let you go, here I pick you up and here I carry you. Imagine how lonely she has been since she was four."* (P8).

Dialysis treatment, especially HD, limits the time women can spend with their children, preventing them from engaging in many activities, especially those outside the home, such as going out to dinner or going to the cinema:

*"My children are bored, because they want to do more things with me, and I can't because I'm on dialysis. I have missed a lot of time, all the time I have been confined to the illness, to the hospital."* (P3).

*"Many times, [the children will say] 'mum can we do, can we go, can. . .' you go out for dinner or to the cinema or whatever and suddenly you have to give up or leave, because you have to go back and connect to your treatment".* (P6).

*"Treatment means being here for 4 hours, 3 days a week I could be with my children somewhere, and I can't because I have to be here [dialysis]."(P10).*

The women described how their children, despite their efforts not to be affected by their illness, were aware of it and often avoided asking their mother directly:

*"I try not to complain too much, for my children's sake. I don't want them to suffer and for them to see me in a bad state. I try to swallow everything I can, many things that happen to me. I try to get them to take it well, but they realize that you feel unwell. They don't ask so that it doesn't hurt them, but they suffer a lot."* (P9).

The women participating in this study related how their children prayed or wished for a transplant for their mother as Christmas presents:

*"At Christmas, there was one year when my daughter's only request was for a transplant for her mum. Up to that point I was aware of everything. She has seen me feeling very ill."* (P2)

*"My children pray to God: 'mum, may you get a new kidney so that you can go out and do things.'"* (P3).

*"It was New Year's Day and at night everyone asked for a kidney for me."* (P11).

Once the children have grown up, many women no longer hide the disease. On the contrary, the children become great supporters and even offer their mother the possibility of becoming their kidney donor. This option is rejected by all women:

*"What I couldn't discuss with my daughter at the time because she was a child, I now discuss with her. I don't hide anything from her. Now, far from feeling lonely, I feel more accompanied."* (P2).

*"My children had all the tests for the transplant, and I said no, they could have all the tests they wanted but none of them would give me their kidney."* (P11).

*"It was when my children were older that I started to tell them. Their response was that they were offering me their kidney. I told them that they are young, I have already lived the life*

*that was meant for me. To take a kidney from one of my children is unacceptable, especially when they are young."* (P14).

## Fear of genetic transmission

There are a wide variety of inherited kidney diseases. Women with CKD express concern that their children may develop the disease:

*"It is a very important problem and although I was told that my problem is not genetic, that it was a consequence of repeated infections, it is always present. You are conditioned, what if I pass it on to my child?"* (P2).

*"I hope that the disease will stop with me, that it won't be passed on to my children. I would not want my children to go through all this again, let alone suffer from it. I hope that when they have their children, this disease will stop. It is my only concern, I am afraid".* (P3).

Women reported sharing this concern with pediatricians, and how they have begun to regularly take their children to check-ups to detect the disease as early as possible:

*"Because when they were born, I said to him: 'doctor, could they develop the disease? What I don't want is for this to happen again, that's the only thing I don't want."* (P3).

*"Thank God, so far there have been no problems, at least from the tests they are undergoing, they are fine. I told the doctor who cares for my children about it and so far, the tests have shown that everything is fine."* (P9).

Fear of passing on the disease and trying to spare their children the suffering that CKD and its treatment entails, leads some women to make the decision to be a mother again from the moment of the diagnosis:

*I haven't had children since. Nor am I going to have them. I decided against it when I was given the diagnosis and told that it could be inherited. I couldn't bear to see a small child with it, especially if it's my child. I have to suffer this but it won't go on [in the family]. I am not going to allow the possibility of giving this to a child."* (P13).

## Discussion

From the moment a woman is diagnosed with CKD, she must undergo a process in which she faces a completely different reality from the one she has experienced until then. A series of important changes are implemented, together with the activation of adaptive mechanisms to the current health situation [25]. The woman is forced to change her lifestyle, changing her diet, frequent contact with the hospital environment and interruption of her daily activities, often affecting her work or studies [26]. These changes in women with ESKD affect the family, their role as the primary caregiver, medication, and the health care system [17, 27].

Gender differences exist in kidney disease [10]. Women have a higher prevalence of specific autoimmune diseases that, together with pregnancy, form a unique challenge for women at risk of CKD and AKI [10]. In addition, there is a lower proportion of women on dialysis than men, and women are less likely to undergo renal transplantation [10, 28]. Advanced-stage CKD most commonly occurs in women after childbearing age. However, the prevalence of advanced stage CKD is increasing and may affect up to 3% of women of childbearing age [29].

Menstrual disturbances as a consequence of endocrine disorders and infertility are common in women with advanced stage CKD, making conception difficult and causing significant disruptions in family planning [30]. In our study, none of them mentioned having discussed fertility issues, the risk of inheritance of the disease, or the risks of pregnancy with their doctors. Lewis & Arber [31] describe how women affected by advanced stage CKD are desperate to have a baby and stop taking "precautions", ignore medical advice to wait until the arrival of a new kidney and even say that if someone advised them that they could not have children, it would be the breaking point for them. Others prefer adoption to risking their own health and that of their baby [31]. In addition to the strong desire for being mothers, these women experience guilt for not meeting social expectations, fear of birth defects and fetal damage, emotional conflict during decision making and fears of exacerbation of the disease [14]. Tong et al., describe how women with advanced stage CKD report feelings of loss when denied motherhood, guilt about disappointing partners and family members, guilt about gambling with their health, and rationalising the health risks of pregnancy in order to pursue motherhood [14].

Women with CKD should be informed about the possibilities of pregnancy and associated risks, including fetal outcomes and maternal complications [32]. Special attention should be given to women with advanced stage CKD of childbearing age, especially young women, for whom pregnancy and childbirth are particularly challenging, helping them to deal with their reproductive dilemmas, whether they should have children, and trying to meet their reproductive needs [33]. At older ages, most women understand that the gradual loss of reproductivity is part of the "normal" ageing process [34]. The importance of pre-pregnancy counseling in women with ESKD needs to be highlighted, and the challenges that may be faced during childbearing discussed. Preconception counselling by multidisciplinary specialized teams is essential to support women in making informed, autonomous decisions, and to maximize opportunities to modify medical care to optimize outcomes if pregnancy occurs [35]. Counseling should be provided to women with ESKD and their partners at the onset of the disease, discussing aspects such as the possibility of delaying pregnancy (contraceptive use), timing of conception, control of renal disease, previous episodes of preeclampsia or AKI injury in pregnancy, possibility of prematurity of the children, possibility of low birth weight, delaying pregnancy until obtaining kidney transplantation, and the possibility of a low birth weight [10, 17, 35–37]. Piccoli et al. [38], in their study on improving care in Kidney Diseases and Pregnancy, describe how it is necessary for the clinical approach to include ethical aspects of pregnancy in patients with diabetes, hypertension and immunological diseases, and the importance of taking into consideration the different cultural and religious contexts. Currently, there are still unresolved issues that women need to be informed about, such as the role of urinary tract infections, kidney stones and urinary malformations [38].

Regarding the presence of children, the main concern of women with ESKD is the disruption of mother/caregiver roles. The role of women as mothers is deeply disrupted as they feel that they neglect their children. Their limited physical abilities, lack of energy and other negative emotions associated with dialysis days strongly alter their roles as mothers, going from being a "strong" woman to a more vulnerable role [39]. Haemodialysis treatment not only has effects on parenting but also on their role as primary providers and support systems for their families. For these women, the inability to continue in their roles seems to influence them to feel a sense of loss of "who I am as a woman" [40].

The women in our study did not report the presence of conflicts between caring for their children and prioritizing their own health needs. One possible explanation for this is that our participants prioritized their children over their health, in an attempt to avoid their illness conditioning and affecting the life and future of their children at all costs. Piccoli et al. [36], described that women undergoing dialysis treatment had low levels of stress due to a

phenomenon referred to by these authors as 'positive defense' of their children. In addition, Wadd et al. [17], reported how women in ESKD with dialysis treatment struggle to care for their children and stay connected to family life, above their own health needs. These authors [17] show similar results to ours, since mothers with ESKD were reluctant to talk about their illness, particularly with their children, even if the child was scared their mother would die. They continued to present a veneer of normality, trying to stay connected with the family, solving all the problems, and attempting to prevent her illness from entering her children's lives. Moreover, Piccoli et al. [36], identified that a robust family network was essential for reducing the impact on the psychological wellbeing of the offspring.

This study has limitations concerning generalizability. Also, gender roles should be considered when considering concealment of disease, as gender may underlie differences in disease experience and how it affects daily life, perspectives of pregnancy, fears for maternal, and foetus health, decision-making insecurities and conflicts regarding maternity and autonomy [19]. Also, it is important to consider the perspectives of the patients' partners for a more comprehensive analysis of the impact of dialysis on motherhood. Finally, the present study has not sampled the different types of renal replacement therapy. It would be necessary to develop new studies describing the perspective of women with ESKD, focusing specifically on each of the different types of treatment. Another limitation is that the participants included in the study are patients with a high average age (57,5 years). Treatment and medical care have changed in the area of obstetric nephrology over the last 20 years [10, 37]. Intensive hemodialysis is now the standard model of care for women receiving dialysis in pregnancy, particularly if residual renal function is minimal [41–43]. Thus, in women with ESKD, intensification of hemodialysis, targeting a serum blood urea nitrogen <35 mg/dL or 36 hours of dialysis per week in women with no residual kidney function, is associated with improved live birth rates and longer gestational age [43]. Hladunewich [42], and Oliverio & Hladunewich [43] concluded that pregnancy may be safe and feasible in women with ESKD receiving intensive hemodialysis. Moreover, fertility is significantly increased after kidney transplantation, and the chances of successful delivery are increased compared to women on dialysis [20]. All these changes may influence the results of the present study because there may be different perspectives between older and younger patients about the disease and its treatment [10, 37]. In the present study, no specific questions were asked to determine the perspective regarding treatment and medical care between older and younger women. In future studies, it would be necessary to include this criterion to compare and describe the perspectives and expectations regarding renal replacement therapy among women with different age groups, for example, adolescents (13–19 years), adults (20–64 years) and older women (>65 years).

## Conclusions

Women with ESKD who require RRT often experience difficulties and challenges in coping with motherhood, struggling to prevent their children from being affected by their disease and expressing great concerns about the possible genetic transmission of the disease. These findings are relevant to women requiring long-term dialysis. Our results provide insight on how motherhood, and childcare are experienced by women with ESKD and may be helpful in dealing with women who suffer from dialysis treatment, and follow-ups. There is a need to develop programs that integrate health and social interventions that help women with ESKD to be able to better integrate their family life with the treatment of the disease. Finally, it should be noted that becoming pregnant, and having children while on dialysis is a major challenge for women with ESKD that has risks, however, pre-pregnancy counselling provides an opportunity to learn about the risks and challenges and enables women and their partners to make an

informed decision and discuss their doubts and fears, and choose different options (assisted fertility, adoption, surrogacy). In addition, intensive dialysis has resulted in a greater likelihood of a better pregnancy outcome, and kidney transplantation can restore a woman's fertility.

## Supporting information

**S1 Interview guide review.**
(DOCX)

## Acknowledgments

We thank the participants for their kind collaboration and participation in this research study.

## Author Contributions

**Conceptualization:** Miriam Álvarez-Villarreal, Juan Francisco Velarde-García, Pilar Carrasco-Garrido, Domingo Palacios-Ceña.

**Data curation:** Miriam Álvarez-Villarreal, Juan Francisco Velarde-García, Cristina García-Bravo, Pilar Carrasco-Garrido, Carmen Jimenez-Antona, Paloma Moro-Lopez-Menchero, Domingo Palacios-Ceña.

**Formal analysis:** Miriam Álvarez-Villarreal, Juan Francisco Velarde-García, Domingo Palacios-Ceña.

**Investigation:** Miriam Álvarez-Villarreal, Juan Francisco Velarde-García, Cristina García-Bravo, Pilar Carrasco-Garrido, Carmen Jimenez-Antona, Domingo Palacios-Ceña.

**Methodology:** Miriam Álvarez-Villarreal, Juan Francisco Velarde-García, Cristina García-Bravo, Pilar Carrasco-Garrido, Carmen Jimenez-Antona, Paloma Moro-Lopez-Menchero, Domingo Palacios-Ceña.

**Resources:** Cristina García-Bravo, Pilar Carrasco-Garrido, Carmen Jimenez-Antona, Paloma Moro-Lopez-Menchero.

**Supervision:** Juan Francisco Velarde-García, Domingo Palacios-Ceña.

**Validation:** Cristina García-Bravo, Pilar Carrasco-Garrido, Carmen Jimenez-Antona, Paloma Moro-Lopez-Menchero.

**Writing – original draft:** Miriam Álvarez-Villarreal, Domingo Palacios-Ceña.

**Writing – review & editing:** Juan Francisco Velarde-García, Pilar Carrasco-Garrido, Domingo Palacios-Ceña.

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
