## [Decision Letter · Decision Letter 0]

10 Jun 2021

PONE-D-21-15320

The experience of motherhood with chronic kidney disease: A qualitative study of women receiving treatment at an ambulatory dialysis unit.

PLOS ONE

Dear Dr. García,

Thank you for submitting your manuscript to PLOS ONE. After careful consideration, we feel that it has merit but does not fully meet PLOS ONE’s publication criteria as it currently stands. Therefore, we invite you to submit a revised version of the manuscript that addresses the points raised during the review process.

Both expert reviewers have expressed interest in the study and MS, and highlight novelties described. The relevance of this study is not limited to nephrologists and dialysis teams, and extends to patients and their families. It is clear from below comments that authors should provide more clarity on the study population (and and highlight these were all women with ESKD on dialysis and not just CKD, and Tx pts were not included), the introduction should be shortened, and some of the research outcomes could be more focused as per reviewers' advice. Please revise the MS thoroughly according to suggestions as per below, and kindly answer to every comment in a point-by-point fashion. The revised MS will undergo thorough peer review again, and there is no guaranteed acceptance after revisions.

We look forward to receiving your revised manuscript.

Kind regards,

Frank JMF Dor, M.D., Ph.D., FEBS, FRCS

Academic Editor

PLOS ONE

Journal Requirements:

2. Please provide additional details regarding participant consent. In the ethics statement in the Methods and online submission information, please ensure that you have specified : 1) whether the ethics committee approved the verbal/oral consent procedure, 2) why written consent could not be obtained, and 3) how verbal/oral consent was recorded. If your study included minors, please state whether you obtained consent from parents or guardians in these cases. If the need for consent was waived by the ethics committee, please include this information.

3. Please include a copy of the interview guide used as a Supporting Information file.

Reviewers' comments:

Reviewer's Responses to Questions

**Comments to the Author**

1. Is the manuscript technically sound, and do the data support the conclusions?

Reviewer #1: Yes

Reviewer #2: Yes

2. Has the statistical analysis been performed appropriately and rigorously? 

Reviewer #1: N/A

Reviewer #2: N/A

3. Have the authors made all data underlying the findings in their manuscript fully available?

Reviewer #1: Yes

Reviewer #2: Yes

4. Is the manuscript presented in an intelligible fashion and written in standard English?

Reviewer #1: Yes

Reviewer #2: Yes

5. Review Comments to the Author

Reviewer #1: Thank you for the opportunity to review this interesting manuscript, which explores the experience of motherhood for women with ESKD.

Unfortunately I do not have expertise in qualitative research, but to my limited knowledge the methodology is appropriate and reporting robust. I have a few suggestions about how to improve the manuscript.

I think that the focus of the study findings should be on motherhood rather than pregnancy. The novelty of this work is describing the challenges of being a parent on dialysis. I suggest that women who do not have pregnancies are not included, as they are not aligned with the study aim.

The authors need to make it clearer that these findings are relevant to women requiring long-term dialysis. They should highlight that all women had ESKD not just CKD, and women with transplants were excluded. I think the inclusion criteria could be clearer.

The introduction is too long and lacks focus – I suggest starting at Page 4 Line 12.

I was interested to note that a conflict between caring for offspring and prioritising own health needs was not identified. In my experience caring for women on HD, they frequently have to make choices about their health care which may compromise their children or vice versa. Did these women have good family support which circumvented this issue. Could current social situation / and/or situation during child-rearing years be described. Giorgina Piccoli identified that a robust family network was essential for reducing impact on offspring psychological wellbeing – this study should be discussed.

What were the mechanisms that women had to use to ensure that their children were well cared for?

What were the ages of children – currently and at time of starting dialysis.

Why was the sample selected according to the research question rather than to represent different RRT modalities, and vintage?

Page 4; line 24 -qualitative research is important to all health care professionals not just nurses!

I am unclear about how losses and limitations regarding driving, working etc is relevant to the research question – could this be discussed more in the context of motherhood – or left-out as dietary restrictions and travel is not just specific to mothers.

The discussion is succinct and well written, but should include more detail about the challenge of not accepting a kidney from offspring, and the importance of discussing motherhood challenges in pre-pregnancy counselling.

Reviewer #2: This manuscript highlights the important difficulties that women face when embarking on motherhood whilst on dialysis.

I am not personally experienced with the methodology described although it would appear comprehensive.

The title and introduction of the manuscript need to clearly state that this refers to women with end-stage kidney disease on dialysis. The importance of this distinction being that the challenges are indeed far greater in this cohort of women than in those with mild chronic kidney disease.

Page 3 line 15 to 17 - I do not understand - are the authors saying that CKD is the fifth most common cause of kidney failure after diabetes, hypertension, glomerulonephritis and pyelonephritis? These diagnoses all cause CKD! This should be deleted.

I note the mean age of the women was 54.1yrs - how old are the children / how long ago were they born? Medical care has changed hugely in the are of obstetric nephrology over the last 30 yrs - could the authors comment on whether there was a difference in the challenges faced between the older and younger cohort of women?

Conceiving / successful pregnancy / motherhood whilst on dialysis is of course hugely challenging and the themes in this paper are important to share with patients and healthcare professionals. However, it would be worthwhile giving a message of hope:

1. Prepregnancy counselling should be promoted as it gives the opportunity to inform of these risks and challenges. It empowers women and their partners in decision making and gives them the opportunity to discuss their fears. It is also an opportunity to optimise risk factors and highlight options (assisted fertility, adoption, surrogacy).

2. The data from Michelle Hladunewich and Luders et al has demonstrated intensive dialysis prescription gives a higher chance of a better pregnancy outcome and this has changed our practice in the last 5-10 years.

3. A well functioning transplant restores fertility and reduces that ESKD on dialysis poses to a pregnancy.

4. Greater awareness of the challenges faced paves the way to greater support for these women.

6. PLOS authors have the option to publish the peer review history of their article (what does this mean?). If published, this will include your full peer review and any attached files.

Reviewer #1: No

Reviewer #2: No

---

## [Author Response · Author response to Decision Letter 0]

16 Jul 2021

RESPONSE LETTER PONE-D-21-15320

Decision on Manuscript PONE-D-21-15320

Entitled: The experience of being a mother with end stage renal disease: A qualitative study of women receiving treatment at an ambulatory dialysis unit

Journal: PLOS ONE 

We would like to thank the Editors and the Reviewers for their careful consideration of our manuscript. We would also like to thank the Reviewers’ suggestions, which we believe have enhanced the quality of the manuscript. We have highlighted (in yellow) all the changes we have made throughout the text. Below please find a detailed list of how we have addressed each comment.

Reviewers' Comments to Author:

REVIEWER: 1

Thank you for the opportunity to review this interesting manuscript, which explores the experience of motherhood for women with ESKD. Unfortunately I do not have expertise in qualitative research, but to my limited knowledge the methodology is appropriate and reporting robust. 

Response: Thank you very much for your comments. We truly appreciate them.

I think that the focus of the study findings should be on motherhood rather than pregnancy. The novelty of this work is describing the challenges of being a parent on dialysis. I suggest that women who do not have pregnancies are not included, as they are not aligned with the study aim.

Response: We have followed the reviewer’s recommendations. We have removed participants who did not have pregnancies.

Also, we have edited the wording throughout the manuscript, and re-written the paragraphs regarding the methods and results sections.

The authors need to make it clearer that these findings are relevant to women requiring long-term dialysis. 

Response: We have followed the reviewer´s recommendations. We have addressed this in the abstract section, and the conclusion section of the main manuscript. 

They should highlight that all women had ESRD not just CKD, and women with transplants were excluded. I think the inclusion criteria could be clearer.

Response: Thank you very much for your comments. 

New information on end stage renal disease, its relationship to CKD, and renal replacement therapies is included in the introduction.

In addition, new information on inclusion criteria is included. As the aim of this study is to describe the experience of parenting and caregiving in women with ESRD, there may be transplanted women with ESRD who can provide relevant and key information for this study. That was the reason why we included women who may have had renal transplantation.

The introduction is too long and lacks focus – I suggest starting at Page 4 Line 12.

Response: We have followed the reviewer’s recommendations. We have shortened the introduction. Moreover, the authors believe that although we reduce the introduction, it is necessary to keep some key parts of the article, because the journal PLOS One is not specialized in nephrology, and some key information such as defining what is chronic kidney disease, end stage renal disease, giving some data on its magnitude, and clarifying or defining treatments would help readers to better understand the article. Plos One journal (https://journals.plos.org/plosone/s/journal-information#loc-scope) showed that: “ PLOS ONE is an inclusive journal community working together to advance science for the benefit of society, now and in the future. Founded with the aim of accelerating the pace of scientific advancement and demonstrating its value, we believe all rigorous science needs to be published and discoverable, widely disseminated and freely accessible to all. The research we publish is multidisciplinary and, often, interdisciplinary. PLOS ONE accepts research in over two hundred subject areas across science, engineering, medicine, and the related social sciences and humanities. We evaluate submitted manuscripts on the basis of methodological rigor and high ethical standards, regardless of perceived novelty.”

I was interested to note that a conflict between caring for offspring and prioritising own health needs was not identified. In my experience caring for women on HD, they frequently have to make choices about their health care which may compromise their children or vice versa. Did these women have good family support which circumvented this issue. Could current social situation / and/or situation during child-rearing years be described. Giorgina Piccoli identified that a robust family network was essential for reducing impact on offspring psychological wellbeing – this study should be discussed.

Response: In our results, the women who participated did not report the presence of conflicts between caring for their children and prioritizing their own health needs. The authors believe that this could be explained by the fact that for them the priority was their children and they were focused on the disease not affecting them, and above all not interfering with their lives. The authors have found previous studies (Wadd et al., 2014) that point to results similar to our findings, where mothers with ESRD under dialysis treatment try to continue to resolve the family's day-to-day life struggle so that the disease does not interfere with their children's lives, making their children and family a priority.

We have included the following text in the discussion section:

The women in our study did not report the presence of conflicts between caring for their children and prioritizing their own health needs. One possible explanation for this is that our participants prioritized their children over their health, in an attempt to avoid their illness conditioning and affecting the life and future of their children at all costs. Piccoli et al [36] described that women undergoing dialysis treatment had low levels of stress due to a phenomenon referred to by these authors as 'positive defense' of their children. In addition, Wadd et al [17] reported how women in ESRD with dialysis treatment struggle to care for their children and stay connected to family life, above their own health needs. These authors [17] show similar results to ours, since mothers with ESRD were reluctant to talk about their illness, particularly with their children, even if the child was scared their mother would die. They continued to present a veneer of normality, trying to stay connected with the family, solving all the problems, and attempting to prevent her illness from entering her children's lives. Moreover, Piccoli et al [36] identified that a robust family network was essential for reducing the impact on the psychological wellbeing of the offspring. 

Additionally, we have included new references:

• Piccoli GB, Postorino V, Cabiddu G, Ghiotto S, Guzzo G, Roggero S, Manca E, Puddu R, Meloni F, Attini R, Moi P, Guida B, Maxia S, Piga A, Mazzone L, Pani A, Postorino M; 'Kidney and Pregnancy Study Group' of the 'Italian Society of Nephrology'. Children of a lesser god or miracles? An emotional and behavioural profile of children born to mothers on dialysis in Italy: a multicentre nationwide study 2000-12. Nephrol Dial Transplant. 2015 Jul;30(7):1193-202. doi: 10.1093/ndt/gfv127. 

• Wadd KM, Bennett PN, Grant J. Mothers requiring dialysis: parenting and end-stage kidney disease. J Ren Care. 2014 Jun;40(2):140-6. doi: 10.1111/jorc.12066. 

What were the ages of children – currently and at time of starting dialysis.

Response: The authors have now included the age of the children at the start of renal replacement therapy. We would like to point out that the study is based on describing women's experience of parenting and child care. Childcare ranges from the care of newborns of mothers with ESRD who have had children during their renal replacement therapy to the care of older children who are part of the family and who live with the woman with ESRD, and where they may be influenced by having a mother with ESRD.

We included a new table 2.

Table 2. Data on renal replacement therapy, and age of the children at the start of treatment.

Participant Age Type

RRTa Initial

RRT Time since CKDe diagnosis until beginning of RRT

(years) nº

offspring Age of children

at 

start

of treatment 

(years)

 Children who

 lived in the 

home of the woman 

with ESRDf Partner

1 63 HDb 1993 < 1 1 11 Yes No.

Separated

2 33 HD 2011 3 2 1 and 3 Yes Yes

3 67 HD 2001 7 4 Not available Not available Yes

4 54 HD 2014 < 1 6 13, 18, 23, 26, 32 and 34 Yes Yes

5 64 HD 2015 < 1 2 39 and 42 No No

6 47 PDc 2014 9 2 13 and 15 Yes Yes

7 50 HD 2017 5 1 21 Yes Yes

8 56 PD 2017 16 1 30 Yes No.

Separated

9 64 TxRd 2015 5 2 34 and 38 No Yes

10 46 HD 2017 < 1 2 21 and 23 Yes Yes

11 69 HD 2000 < 1 5 15, 25, 27, 29 and 31 Yes Yes

12 72 HD 2014 < 1 1 42 No Yes

13 42 HD 2017 < 1 1 16 Yes Yes

14 78 HD 2017 39 2 Not available Not available Yes

aRRT: renal replacement therapy; bHD: hemodialysis; cPD: peritoneal dialysis; dTxR: kidney transplant; eCKD: chronic kidney disease; fESRD: end-stage renal disease. 

We have also included the following new text in the introduction section:

In addition, women feel a responsibility towards the health of their partners, children, and other family members [15]. Maternity is associated with several common fears, including the transmission of hereditary kidney disease to children and the inability to survive or be well enough to care for children [16]. In addition to managing their RRT, women with ESRD under dialysis treatment try to focus on fitting everything in, maintaining ties between family members, remaining present in the lives of their children and partner, and not losing the connection to their family and social life, and striving for normality [17].

The data collection section has been edited as follows:

During the interview, at the women’s request, it was clarified that when they were asked about the care of their children during RRT, this included both the care provided to the infants and the daily care that the woman provided to the remaining children she already had.

We have included a new reference:

• Wadd KM, Bennett PN, Grant J. Mothers requiring dialysis: parenting and end-stage kidney disease. J Ren Care. 2014 Jun;40(2):140-6. doi: 10.1111/jorc.12066. Epub 2014 Mar 26. PMID: 24674737.

Why was the sample selected according to the research question rather than to represent different RRT modalities, and vintage?

Response: The research question was used to select the participants. In this study it was to describe the experience of parenting and child care in women with ESRD. The focus of the study was women with ESRD, regardless of the type of renal replacement therapy they had.

In qualitative research, it is accepted to include participants who suffer or live the same phenomenon (e.g. a disease) to know their individual experience and perspective (Creswell & Poth, 2018), even if they present differences in other aspects such as treatment, clinical presentation or different degrees of disability (Moser & Korstjens, 2018). Thus in this study we included women with ESRD, who could present with different renal replacement therapies, and they were chosen because they are the participants who are best able to provide relevant information.

Moser & Korstjens (2018) reported that. “Sampling is the process of selecting or searching for situations, context and/or participants who provide rich data of the phenomenon of interest (…) In qualitative research, you sample deliberately, not at random. The most commonly used deliberate sampling strategies are purposive sampling, criterion sampling, theoretical sampling, convenience sampling and snowball sampling (…) Key informants hold special and expert knowledge about the phenomenon to be studied and are willing to share information and insights with you as the researcher.” (p.10)

Furthermore, the authors recognize that sampling on the basis of the different types of treatment could also be a valid and very interesting option. For this reason, limitations include the need to conduct studies describing the perspective of women with ESRKD based on the different treatments.

We included the following in the limitations section:

Finally, the present study has not sampled the different types of renal replacement therapy. It would be necessary to develop new studies describing the perspective of women with ESRD, focusing specifically on each of the different types of treatment.

References:

• Creswell, J.W.; Poth, C.N. Qualitative Inquiry and Research Design. Choosing among Five Approaches, 4th ed.; Sage: Thousand Oaks, CA, USA, 2018

• Moser A, Korstjens I. Series: Practical guidance to qualitative research. Part 3: Sampling, data collection and analysis. Eur J Gen Pract. 2018 Dec;24(1):9-18. doi: 10.1080/13814788.2017.1375091. 

Page 4; line 24 -qualitative research is important to all health care professionals not just nurses!

Response: We agree with the reviewer. We have followed the reviewer´s recommendations and removed this sentence.

I am unclear about how losses and limitations regarding driving, working etc is relevant to the research question – could this be discussed more in the context of motherhood – or left-out as dietary restrictions and travel is not just specific to mothers.

Response: We agree with the reviewer. We have removed theme 4. We have changed it and rewritten the abstract, results and discussion sections.

The discussion is succinct and well written, but should include more detail about the challenge of not accepting a kidney from offspring, and the importance of discussing motherhood challenges in pre-pregnancy counselling.

Response: We have followed the reviewer´s recommendations.

We have included the following in the discussion section. 

The importance of pre-pregnancy counseling in women with ESRD needs to be highlighted, and the challenges that may be faced during childbearing discussed. Preconception counselling by multidisciplinary specialized teams is essential to support women in making informed, autonomous decisions, and to maximize opportunities to modify medical care to optimize outcomes if pregnancy occurs [35]. Counseling should be provided to women with ESRD and their partners at the onset of the disease, discussing aspects such as the possibility of delaying pregnancy (contraceptive use), timing of conception, control of renal disease, previous episodes of preeclampsia or acute kidney injury in pregnancy, possibility of prematurity of the children, possibility of low birth weight, delaying pregnancy until obtaining kidney transplantation, and the possibility of a low birth weight [10,17,35-37]. Piccoli et al [38] in their study on improving care in Kidney Diseases and Pregnancy, describe how it is necessary for the clinical approach to include ethical aspects of pregnancy in patients with diabetes, hypertension and immunological diseases, and the importance of taking into consideration the different cultural and religious contexts. Currently, there are still unresolved issues that women need to be informed about, such as the role of urinary tract infections, kidney stones and urinary malformations [38].

We have included new references:

• Cabiddu G, Spotti D, Gernone G, Santoro D, Moroni G, Gregorini G, Giacchino F, Attini R, Limardo M, Gammaro L, Todros T, Piccoli GB; Kidney and Pregnancy Study Group of the Italian Society of Nephrology. A best-practice position statement on pregnancy after kidney transplantation: focusing on the unsolved questions. The Kidney and Pregnancy Study Group of the Italian Society of Nephrology. J Nephrol. 2018 Oct;31(5):665-681. doi: 10.1007/s40620-018-0499-x. 

• Fitzpatrick A, Mohammadi F, Jesudason S. Managing pregnancy in chronic kidney disease: improving outcomes for mother and baby. Int J Womens Health. 2016 Jul 14;8:273-85. doi: 10.2147/IJWH.S76819. 

• Piccoli GB, Alrukhaimi M, Liu ZH, Zakharova E, Levin A. What We Do and Do Not Know about Women and Kidney Diseases; Questions Unanswered and Answers Unquestioned: Reflection on World Kidney Day and International Women's Day. Kidney Dis (Basel). 2018 Feb;4(1):37-48. doi: 10.1159/000485269. 

• Piccoli GB, Attini R, Cabiddu G. Kidney Diseases and Pregnancy: A Multidisciplinary Approach for Improving Care by Involving Nephrology, Obstetrics, Neonatology, Urology, Diabetology, Bioethics, and Internal Medicine. J Clin Med. 2018 Jun 4;7(6):135. doi: 10.3390/jcm7060135. 

• Piccoli GB, Postorino V, Cabiddu G, Ghiotto S, Guzzo G, Roggero S, Manca E, Puddu R, Meloni F, Attini R, Moi P, Guida B, Maxia S, Piga A, Mazzone L, Pani A, Postorino M; ‘Kidney and Pregnancy Study Group’ of the ‘Italian Society of Nephrology’; 'Kidney and Pregnancy Study Group' of the 'Italian Society of Nephrology'. Children of a lesser god or miracles? An emotional and behavioural profile of children born to mothers on dialysis in Italy: a multicentre nationwide study 2000-12. Nephrol Dial Transplant. 2015 Jul;30(7):1193-202. doi: 10.1093/ndt/gfv127. 

• Wadd KM, Bennett PN, Grant J. Mothers requiring dialysis: parenting and end-stage kidney disease. J Ren Care. 2014 Jun;40(2):140-6. doi: 10.1111/jorc.12066. 

REVIEWER: 2

The title and introduction of the manuscript need to clearly state that this refers to women with end-stage kidney disease on dialysis. The importance of this distinction being that the challenges are indeed far greater in this cohort of women than in those with mild chronic kidney disease.

Response: We have followed the reviewer´s recommendations.

We have included new information to the introduction section.

Page 3 line 15 to 17 - I do not understand - are the authors saying that CKD is the fifth most common cause of kidney failure after diabetes, hypertension, glomerulonephritis and pyelonephritis? These diagnoses all cause CKD! This should be deleted.

Response: Thank you very much for your comments. We have removed the paragraph and rewritten the introduction section. We have also followed the recommendations by Reviewer 1.

I note the mean age of the women was 54.1yrs - how old are the children / how long ago were they born? Medical care has changed hugely in the are of obstetric nephrology over the last 30 yrs - could the authors comment on whether there was a difference in the challenges faced between the older and younger cohort of women?

Response: We agree with the reviewer that the mean age is high and may condition the perspective of the participants, since their experience with the renal treatment received may have been very different. In this study we have not asked specific questions about this aspect, nor have we compared the narratives of patients with different age ranges. The authors believe that this aspect is relevant and it is included and described in the limitations section of the study. In addition, the changes that the authors believe have set the tone in the treatment of women with ESRD who wish to have children are also included.

We have included the following text in the discussion section:

Finally, the present study has not sampled the different types of renal replacement therapy. It would be necessary to develop new studies describing the perspective of women with ESRD, focusing specifically on each of the different types of treatment. Another limitation is that the participants included in the study are patients with a high average age (57,5 years). Treatment and medical care have changed in the area of obstetric nephrology over the last 20 years [10,37]. Intensive hemodialysis is now the standard model of care for women receiving dialysis in pregnancy, particularly if residual renal function is minimal [41 - 43]. Thus, in women with ESRD, intensification of hemodialysis, targeting a serum blood urea nitrogen <35 mg/dL or 36 hours of dialysis per week in women with no residual kidney function, is associated with improved live birth rates and longer gestational age [43]. Hladunewich [42], and Oliverio & Hladunewich [43] concluded that pregnancy may be safe and feasible in women with ESRD receiving intensive hemodialysis. Moreover, fertility is significantly increased after kidney transplantation, and the chances of successful delivery are increased compared to women on dialysis [20]. All these changes may influence the results of the present study because there may be different perspectives between older and younger patients about the disease and its treatment [10,37]. In the present study, no specific questions were asked to determine the perspective regarding treatment and medical care between older and younger women. In future studies, it would be necessary to include this criterion to compare and describe the perspectives and expectations regarding renal replacement therapy among women with different age groups, for example, adolescents (13-19 years), adults (20-64 years) and older women (>65 years). 

Also, we have included new references:

• Cabiddu G, Spotti D, Gernone G, Santoro D, Moroni G, Gregorini G, Giacchino F, Attini R, Limardo M, Gammaro L, Todros T, Piccoli GB; Kidney and Pregnancy Study Group of the Italian Society of Nephrology. A best-practice position statement on pregnancy after kidney transplantation: focusing on the unsolved questions. The Kidney and Pregnancy Study Group of the Italian Society of Nephrology. J Nephrol. 2018 Oct;31(5):665-681. doi: 10.1007/s40620-018-0499-x. Epub 2018 Jun 14. Erratum in: J Nephrol. 2018 Jul 6;: PMID: 29949013; PMCID: PMC6182355.

• Hladunewich MA. Chronic Kidney Disease and Pregnancy. Semin Nephrol. 2017 Jul;37(4):337-346. doi: 10.1016/j.semnephrol.2017.05.005. PMID: 28711072.

• Oliverio AL, Hladunewich MA. End-Stage Kidney Disease and Dialysis in Pregnancy. Adv Chronic Kidney Dis. 2020 Nov;27(6):477-485. doi: 10.1053/j.ackd.2020.06.001. PMID: 33328064; PMCID: PMC7781109.

• Piccoli GB, Alrukhaimi M, Liu ZH, Zakharova E, Levin A. What We Do and Do Not Know about Women and Kidney Diseases; Questions Unanswered and Answers Unquestioned: Reflection on World Kidney Day and International Women's Day. Kidney Dis (Basel). 2018 Feb;4(1):37-48. doi: 10.1159/000485269. Epub 2018 Feb 1. PMID: 29594141; PMCID: PMC5848484.

• Piccoli GB, Cabiddu G, Daidone G, Guzzo G, Maxia S, Ciniglio I, Postorino V, Loi V, Ghiotto S, Nichelatti M, Attini R, Coscia A, Postorino M, Pani A; Italian Study Group “Kidney and Pregnancy”. The children of dialysis: live-born babies from on-dialysis mothers in Italy--an epidemiological perspective comparing dialysis, kidney transplantation and the overall population. Nephrol Dial Transplant. 2014 Aug;29(8):1578-86. doi: 10.1093/ndt/gfu092. Epub 2014 Apr 22. PMID: 24759612.

• Piccoli GB, Minelli F, Versino E, Cabiddu G, Attini R, Vigotti FN, Rolfo A, Giuffrida D, Colombi N, Pani A, Todros T. Pregnancy in dialysis patients in the new millennium: a systematic review and meta-regression analysis correlating dialysis schedules and pregnancy outcomes. Nephrol Dial Transplant. 2016 Nov;31(11):1915-1934. doi: 10.1093/ndt/gfv395. Epub 2015 Nov 27. PMID: 26614270.

Conceiving / successful pregnancy / motherhood whilst on dialysis is of course hugely challenging and the themes in this paper are important to share with patients and healthcare professionals. However, it would be worthwhile giving a message of hope:

1. Prepregnancy counselling should be promoted as it gives the opportunity to inform of these risks and challenges. It empowers women and their partners in decision making and gives them the opportunity to discuss their fears. It is also an opportunity to optimise risk factors and highlight options (assisted fertility, adoption, surrogacy).

2. The data from Michelle Hladunewich and Luders et al has demonstrated intensive dialysis prescription gives a higher chance of a better pregnancy outcome and this has changed our practice in the last 5-10 years.

3. A well functioning transplant restores fertility and reduces that ESKD on dialysis poses to a pregnancy.

4. Greater awareness of the challenges faced paves the way to greater support for these women.

Response: We have followed the reviewer's indications. New information on the beneficial effect of intensive dialysis in pregnancy is included in the discussion, and recent studies by Hladunewich et al and Piccoli et al are cited (see references below).

We have included in the discussion section:

Treatment and medical care have changed in the area of obstetric nephrology over the last 20 years [10,37]. Intensive hemodialysis is now the standard model of care for women receiving dialysis in pregnancy, particularly if residual renal function is minimal [41-43]. Thus, in women with ESRD, intensification of hemodialysis, targeting a serum blood urea nitrogen <35 mg/dL or 36 hours of dialysis per week in women with no residual kidney function, is associated with improved live birth rates and longer gestational age [43]. Hladunewich [42], and Oliverio & Hladunewich [43] concluded that pregnancy may be safe and feasible in women with ESRD receiving intensive hemodialysis. Moreover, fertility is significantly increased after kidney transplantation, and the chances of successful delivery are increased compared to women on dialysis [20]. 

In addition, further information is included in the conclusions, noting, and highlighting that despite the challenges, today women with ESRD are safer and can safely become mothers.

The following has been included in the conclusions section:

Finally, it should be noted that becoming pregnant, and having children while on dialysis is a major challenge for women with ESRD that has risks, however, pre-pregnancy counselling provides an opportunity to learn about the risks and challenges and enables women and their partners to make an informed decision and discuss their doubts and fears, and choose different options (assisted fertility, adoption, surrogacy). In addition, intensive dialysis has resulted in a greater likelihood of a better pregnancy outcome, and kidney transplantation can restore a woman's fertility.

Also, we have included new references:

• Hladunewich MA. Chronic Kidney Disease and Pregnancy. Semin Nephrol. 2017 Jul;37(4):337-346. doi: 10.1016/j.semnephrol.2017.05.005. PMID: 28711072.

• Oliverio AL, Hladunewich MA. End-Stage Kidney Disease and Dialysis in Pregnancy. Adv Chronic Kidney Dis. 2020 Nov;27(6):477-485. doi: 10.1053/j.ackd.2020.06.001. PMID: 33328064; PMCID: PMC7781109.

• Piccoli GB, Cabiddu G, Daidone G, Guzzo G, Maxia S, Ciniglio I, Postorino V, Loi V, Ghiotto S, Nichelatti M, Attini R, Coscia A, Postorino M, Pani A; Italian Study Group “Kidney and Pregnancy”. The children of dialysis: live-born babies from on-dialysis mothers in Italy--an epidemiological perspective comparing dialysis, kidney transplantation and the overall population. Nephrol Dial Transplant. 2014 Aug;29(8):1578-86. doi: 10.1093/ndt/gfu092. Epub 2014 Apr 22. PMID: 24759612.

• Piccoli GB, Minelli F, Versino E, Cabiddu G, Attini R, Vigotti FN, Rolfo A, Giuffrida D, Colombi N, Pani A, Todros T. Pregnancy in dialysis patients in the new millennium: a systematic review and meta-regression analysis correlating dialysis schedules and pregnancy outcomes. Nephrol Dial Transplant. 2016 Nov;31(11):1915-1934. doi: 10.1093/ndt/gfv395. Epub 2015 Nov 27. PMID: 26614270.

The authors believe that the comments made by the reviewers have improved the quality of the manuscript. We hope that the changes made meet all the observations and expectations of the reviewers.

The Authors

---

## [Decision Letter · Decision Letter 1]

23 Aug 2021

PONE-D-21-15320R1

The experience of being a mother with end stage renal disease: A qualitative study of women receiving treatment at an ambulatory dialysis unit

PLOS ONE

Dear Dr. García,

Thank you for submitting your manuscript to PLOS ONE. After careful consideration, we feel that it has merit but does not fully meet PLOS ONE’s publication criteria as it currently stands. Therefore, we invite you to submit a revised version of the manuscript that addresses the points raised during the review process.

ACADEMIC EDITOR: Conditionally accepted provided changes according to reviewer 2's comments are made.

We look forward to receiving your revised manuscript.

Kind regards,

Frank JMF Dor, M.D., Ph.D., FEBS, FRCS

Academic Editor

PLOS ONE

Journal Requirements:

Reviewers' comments:

Reviewer's Responses to Questions

**Comments to the Author**

1. If the authors have adequately addressed your comments raised in a previous round of review and you feel that this manuscript is now acceptable for publication, you may indicate that here to bypass the “Comments to the Author” section, enter your conflict of interest statement in the “Confidential to Editor” section, and submit your "Accept" recommendation.

Reviewer #1: All comments have been addressed

Reviewer #2: (No Response)

2. Is the manuscript technically sound, and do the data support the conclusions?

Reviewer #1: Yes

Reviewer #2: Yes

3. Has the statistical analysis been performed appropriately and rigorously? 

Reviewer #1: Yes

Reviewer #2: Yes

4. Have the authors made all data underlying the findings in their manuscript fully available?

Reviewer #1: Yes

Reviewer #2: Yes

5. Is the manuscript presented in an intelligible fashion and written in standard English?

Reviewer #1: Yes

Reviewer #2: Yes

6. Review Comments to the Author

Reviewer #1: (No Response)

Reviewer #2: Many thanks for the opportunity to re-review this manuscript. The authors have addressed the reviewers' comments.

A few things:

I think this sentence in the abstract conclusion may still be misleading page 3 line 3-4 'These findings are also relevant to

4 women requiring long-term dialysis.' In fact these findings are ONLY relevant to women on dialysis. I would recommend removing the word 'also'. The findings are not relevant to women who have ESKD but are transplanted.

Page 3 line 13 - 14 - I still think that for the causes of CKD, autoimmune disease and genetic renal disease should be included in this list. I realise they are discussed later but they should be mentioned from the outset.

Page 15 line 3 - typo - CDK should read CKD

Recommend that 'ESRD' be changed to 'ESKD' and acute kidney disease by changed to AKI (acute kidney injury) throughout the manuscript in line with current nomenclature.

7. PLOS authors have the option to publish the peer review history of their article (what does this mean?). If published, this will include your full peer review and any attached files.

Reviewer #1: **Yes: **Kate Bramham

Reviewer #2: No

---

## [Author Response · Author response to Decision Letter 1]

26 Aug 2021

Decision on Manuscript PONE-D-21-15320R1

Entitled: The experience of being a mother with end stage renal disease: A qualitative study of women receiving treatment at an ambulatory dialysis unit

Journal: PLOS ONE 

We would like to thank the Editors and the Reviewers for their careful consideration of our manuscript. We would also like to thank the Reviewers’ suggestions, which we believe have enhanced the quality of the manuscript. We have highlighted (in yellow) all the changes we have made throughout the text. Below please find a detailed list of how we have addressed each comment.

Reviewers' Comments to Author:

REVIEWER: 2

A few things:

I think this sentence in the abstract conclusion may still be misleading page 3 line 3-4: 

'These findings are also relevant to4 women requiring long-term dialysis.' In fact these findings are ONLY relevant to women on dialysis. I would recommend removing the word 'also'. The findings are not relevant to women who have ESKD but are transplanted.

Response: We have followed the reviewer´s recommendations and clarify the sentence.

Deciding to become a mother and taking care of children represents a challenge for women with ESKD, coupled with the losses in their lives caused by the disease. These findings are only relevant to women on dialysis. 

Page 3 line 13 - 14 - I still think that for the causes of CKD, autoimmune disease and genetic renal disease should be included in this list. I realise they are discussed later but they should be mentioned from the outs

Response: We have followed the reviewer´s recommendations and included the causes of kidney disease.

A significant percentage of the population suffer from CKD, which is related to highly prevalent phenomena or diseases, such as ageing, arterial hypertension, diabetes, cardiovascular disease, autoimmune disease or genetic renal disease [3].

Page 15 line 3 - typo - CDK should read CKD

Response: We have followed the reviewer´s recommendation and change it.

Gender differences exist in kidney disease [10]. Women have a higher prevalence of specific autoimmune diseases that, together with pregnancy, form a unique challenge for women at risk of CKD and AKI [10]. 

Recommend that 'ESRD' be changed to 'ESKD' and acute kidney disease by changed to AKI (acute kidney injury) throughout the manuscript in line with current nomenclature.

Response: We have followed the reviewer´s recommendations and change it in the manuscript

The authors believe that the comments made by the reviewers have improved the quality of the manuscript. We hope that the changes made meet all the observations and expectations of the reviewers.

The Authors

---

## [Editor Report · Decision Letter 2]

8 Sep 2021

The experience of being a mother with end stage renal disease: A qualitative study of women receiving treatment at an ambulatory dialysis unit

PONE-D-21-15320R2

Dear Dr. García,

We’re pleased to inform you that your manuscript has been judged scientifically suitable for publication and will be formally accepted for publication once it meets all outstanding technical requirements.

Kind regards,

Frank JMF Dor, M.D., Ph.D., FEBS, FRCS

Academic Editor

PLOS ONE
---

## [Editor Report · Acceptance letter]

13 Sep 2021

PONE-D-21-15320R2 

The experience of being a mother with end stage renal disease: A qualitative study of women receiving treatment at an ambulatory dialysis unit 

Dear Dr. Velarde-García:

I'm pleased to inform you that your manuscript has been deemed suitable for publication in PLOS ONE. Congratulations! Your manuscript is now with our production department. 

Kind regards, 

on behalf of

Dr. Frank JMF Dor 

Academic Editor

PLOS ONE